

# Blast from the past: Constraints on the dark sector from the BEBC WA66 beam dump experiment

**Giacomo Marocco ⋆ and Subir Sarkar**

Rudolf Peierls Centre for Theoretical Physics, University of Oxford,
Parks Road, Oxford OX1 3PU, United Kingdom

⋆ giacomo.marocco@physics.ox.ac.uk

## Abstract

We derive limits on millicharged dark states, as well as particles with electric or magnetic dipole moments, from the number of observed forward electron scattering events at the Big European Bubble Chamber in the 1982 CERN-WA-066 beam dump experiment. The dark states are produced by the 400 GeV proton beam primarily through the decays of mesons produced in the beam dump, and the lack of excess events places bounds extending up to GeV masses. These improve on bounds from all other experiments, in particular CHARM II.

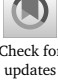

## 1 Introduction

Dark matter, although weakly interacting with the Standard Model, may couple directly to photons, e.g. in theories with kinetic mixing of our photon with a paraphoton [1]. If the para-

photon is massless, the model accommodates particles with a millicharge (mQ), allowing an apparent circumvention of charge quantisation. Alternatively, the coupling may be through operators of mass dimension higher than four; the dimension-five operators involving two fermions and a photon are magnetic dipole moments (MDMs) and electric dipole moments (EDMs). From an effective field theory (EFT) point of view, these operators generically constitute the three most relevant couplings of a dark Dirac fermion $\chi$ to photons.

The question of the existence of millicharged fermions has received far more attention than that of dark states with EDMs or MDMs (see [2,3] for recent reviews). A dedicated search for millicharges was carried out at SLAC, placing limits in the mass range $0.1-100$ MeV [4]; data from LSND [5] and MiniBooNE [6] has been used to place similar constraints at the upper edge of this mass range [7]. Recently it was proposed [8] to use liquid argon detectors to carry out a search, which was performed by the ArgoNeuT experiment, yielding bounds extending into the GeV range [9]. The prototype detector of the milliQan experiment at the LHC has placed new exclusions at masses around 1 GeV [10]. OPAL at LEP [11] probed larger masses up to 100 GeV [12], while bounds from CMS at LHC extend all the way up to a TeV [13]. There are also astrophysical bounds: supernovae constrain states with masses $m_\chi \lesssim 100$ MeV [14], while stellar cooling constrains still lighter masses $m_\chi \lesssim 100$ keV [15].

New states with dipole moments have previously been considered mainly in the context of dark matter [16–20]. More recently, there have been efforts to bound these operators without making any assumptions concerning their relic abundance. Data from L3 at LEP II [21] places strong limits [22], as do high intensity fixed-target experiments [23]. The latter will be the focus of this work.

Past beam dump experiments strongly constrain the photon coupling to any possible new degrees of freedom [23]. In many cases, however, the backgrounds of these experiments are poorly understood, and the corresponding bounds therefore fold in uncertain assumptions. Data from the Big European Bubble Chamber (BEBC) WA66 beam dump experiment [24] was however used some years ago to carry out a *dedicated* search for a MDM of the tau neutrino [25]. Earlier data from the same experiment had been used to set bounds on light gluinos [26] and heavy neutral leptons [27]. Very recently, this data has also been used to bound a model with a heavy dark photon from consideration of a limited number of meson decays [28]. Nevertheless the BEBC WA66 experiment appears to have been largely forgotten in the recent revival of interest concerning the dark sector, even though its sensitivity is still competitive as we will demonstrate below.

In this work we recast the BEBC data in terms of the minimal model of electromagnetic interactions arising from a massless dark photon, and more generally in terms of EDMs and MDMs. In these cases the parameter space is spanned simply by the mass $m_\chi$ of the electromagnetically interacting Dirac fermion $\chi$ and the coupling constant $Q_\chi = \epsilon e$, $d$, or $\mu$ for mQs, EDMs or MDMS, respectively. These parameters enter the Lagrangian through

$$\mathcal{L}_{\text{int}} = \epsilon e A_\mu \bar{\chi} \gamma^\mu \chi + \frac{1}{2}\mu F_{\mu\nu}\bar{\chi}\sigma^{\mu\nu}\chi + \frac{i}{2}d F_{\mu\nu}\bar{\chi}\sigma^{\mu\nu}\gamma^5 \chi, \tag{1}$$

where $\sigma^{\mu\nu} \equiv i[\gamma^\mu, \gamma^\nu]/2$; this Lagrangian is written in a basis where the photon-dark photon system is diagonal. We express dipole moments in units of the Bohr magneton $\mu_B = e/2m_e$, use rationalised units such that $\epsilon_0 = 1$, and consider a situation in which couplings are turned on one at a time.

## 2 BEBC

The operating principle of fixed-target proton experiments is simple. A large number of secondaries are created when a beam of protons is directed on a dump. The length of the dump

Table 1: The relevant experimental parameters for BEBC and CHARM II, and those projected for SHiP. POT is the total number of protons on target, either actual or predicted. $E_b$ is the energy of proton beam. $D$ is the distance from the end of the target to the beginning of the detector. $n_e$ is the detector's electron density; in the case of CHARM II and SHiP, an effective density is given to account for their active layers. $V$ is the detector volume written as transverse area × length; the dimensions of BEBC are given approximating the detector as a cuboid. Cuts are placed on the kinetic energy of the electron $E_e$ and on the angle $\theta_e$ between the beam axis and recoil electron. The number of observed events is given for BEBC and CHARM II, and the expected background for SHiP. The detection efficiency after cuts is denoted $\eta$.

| Experiment | POT/$10^{18}$ | $E_b$/GeV | $D$/m | $n_e/10^{23}$ cm$^{-3}$ | $V$/cm$^3$ | Cuts | Observed | $\eta$ |
|---|---|---|---|---|---|---|---|---|
| BEBC [24,25] | 2.72 | 400 | 404 | 2.6 | $357 \times 252 \times 185$ | $E_e > 1$ GeV $\wedge E_e \theta_e^2 < 2m_e$ | 1 | 0.8 |
| CHARM II [29,30] | 25 | 450 | 870 | 4.3 | $370 \times 370 \times 3567$ | $E_e \in [3, 24]$ GeV $\wedge E_e \theta_e^2 < 2m_e$ | $5429 \pm 120$ | 0.57 |
| SHiP (proposed) [31] | 200 | 400 | 56.5 | 19 | $187 \times 69 \times 87$ | $E_e \in [1, 20]$ GeV $\wedge \theta_e \in [10, 20]$ mrad | 284 (forecast) | 0.5 |

is critical for the type of particles produced. For a thin (beryllium) target, such as that used for CHARM II [29], the dominant production channel of sufficiently light charged particles is charged pion decay. This is also the main source of the conventional background of neutrinos. When however a thick (copper) target is used as for BEBC [24], these charged pions are absorbed before decaying since the mean interaction time is shorter than their lifetime.

The dark states of interest here, coupling only to photons, are mainly produced by scalar mesons that decay into photons, e.g. neutral pions, or heavy vector mesons that usually decay into $\ell^+\ell^-$ pairs. Given the short lifetime of these mesons, such dark states may still be produced in the thick target of BEBC. Any particles produced with a sufficiently weak coupling then traverse some intervening distance, and may then scatter off electrons in a detector downstream of the dump to leave an observable signal.

The BEBC detector was 404 m downstream of a 400 GeV proton beam from the CERN SPS dumped onto a solid copper block [24]. A total of $2.72 \times 10^{18}$ protons on target were accumulated over the experiment. The detector itself had a fiducial volume of 16.6 m$^3$ filled with a neon-hydrogen mixture. A dedicated search for elastically scattered final state electrons was carried out, with one candidate event observed [25]. Relevant details of the BEBC WA66 experiment are given in Table 1 as well as those of CHARM II, along with the proposed SHiP for comparison.

## 3 Dark state signatures at beam dumps

In this section, we detail the calculation of the number of dark states entering the detector and their subsequent scattering. We focus on the dominant production channels, which are meson decays, with Drell-Yan forming a highly subdominant component. The dark state flux is handled primarily by the MADGRAPH plugin MADDUMP [32,33], which provides the distribution of scattered electrons in the detector differential in both energy and angle. The following procedures were validated by reproducing the total number of electron scatterings due to the Standard Model interactions of the neutrino flux measured by CHARM II.

Table 2: The number of mesons produced per proton-proton collision at the relevant energies, using SoftQCD.

| $E_{\rm b}/$GeV | $\pi^0$ | $\eta$ | $\omega$ | $\rho$ | $\phi$ | $J/\psi$ |
|---|---|---|---|---|---|---|
| 400 | 4.0 | $4.6 \times 10^{-1}$ | $5.3 \times 10^{-1}$ | $5.3 \times 10^{-1}$ | $1.9 \times 10^{-2}$ | $6.4 \times 10^{-6}$ |
| 450 | 4.2 | $4.7 \times 10^{-1}$ | $5.5 \times 10^{-1}$ | $5.6 \times 10^{-1}$ | $2.0 \times 10^{-2}$ | $6.6 \times 10^{-6}$ |

### 3.1 Meson decays

The number of dark matter particles $N_\chi$ produced in neutral meson decays is given by

$$N_\chi = 2N_{\rm POT} \sum_{\mathfrak{m}} N_{\mathfrak{m}/{\rm POT}} {\rm Br}(\mathfrak{m} \to \chi \bar\chi + \text{anything}), \tag{2}$$

where $N_{\rm POT}$ is the number of protons on target (POT), and $N_{\mathfrak{m}/{\rm POT}}$ is the number of a particular meson $\mathfrak{m}$ produced per POT. Meson production can be approximated from fixed-target $pp$ collisions simulated in PYTHIA 8.3 [34,35], ignoring for simplicity the fraction of production in hadronic cascades. We then scale cross sections according to the nucleon number of the target $A$ to some power. In reality, this scaling index depends on the kinematics of the process, since mesons can re-interact within a single large nucleus and produce softer secondary products, but when approximating the target as a dilute gas we stipulate a scaling of $A^{2/3}$. The number of mesons we thus estimate to be produced per $pp$ collision are listed in Table 2.

The meson decay into DM is characterised by the branching fraction; parity invariance restricts the decay of the pseudoscalars $\mathfrak{s} = \pi^0, \eta$, while the small value of $\alpha$ implies that

$$\begin{aligned}
{\rm Br}(\mathfrak{s} \to \chi \bar\chi + \text{anything}) &\simeq {\rm Br}(\mathfrak{s} \to \chi \bar\chi \gamma), \\
{\rm Br}(\mathfrak{v} \to \chi \bar\chi + \text{anything}) &\simeq {\rm Br}(\mathfrak{v} \to \chi \bar\chi),
\end{aligned} \tag{3}$$

where we include vector mesons $\mathfrak{v} = \rho, \phi, J/\psi$. For $\omega$ mesons, we find that for light millicharged dark states there is also a significant contribution from the decay into a neutral pion and a dark pair, as detailed in Appendix A. Hence we have

$$\text{mQ}: \quad {\rm Br}(\omega \to \chi \bar\chi + \text{anything}) \simeq {\rm Br}(\omega \to \chi \bar\chi) + {\rm Br}(\omega \to \chi \bar\chi \pi^0), \tag{4}$$

$$\text{MDM, EDM}: \quad {\rm Br}(\omega \to \chi \bar\chi + \text{anything}) \simeq {\rm Br}(\omega \to \chi \bar\chi). \tag{5}$$

The value of the branching ratios can be related to measured Standard Model branching ratios after calculating the respective rates, as outlined in Appendix A.

Although the overall normalisation of the dark state flux depends only on the branching ratios, to determine the kinematic properties requires a more detailed analysis. First, the angular and energy distribution of the meson flux is needed. One possibility is to use experimentally measured distributions. However for neutral pions, this distribution is highly uncertain due to the difficulty of the measurement. Previous works have chosen to invoke isospin invariance to treat the neutral pion distribution as the average of those for charged pions [23,36]. However, since the charged pions are much longer lived, one expects the neutral pions to be scattered less within the target. The heavier mesons tend to have smaller momenta and thus to be more widely distributed, and so are unlikely to follow the same distribution as pions. We thus choose to specify the distribution instead using the full information obtained from PYTHIA. The $\chi$ distributions differential in angle and energy and shown in Figs 1 and 2, respectively.

To calculate the dark state spectrum, we use the Monte Carlo techniques implemented by MADDUMP [33]. This programme takes the meson spectrum as an input and outputs the dark state distribution using an EFT framework for the interactions. In the case of pseudoscalar

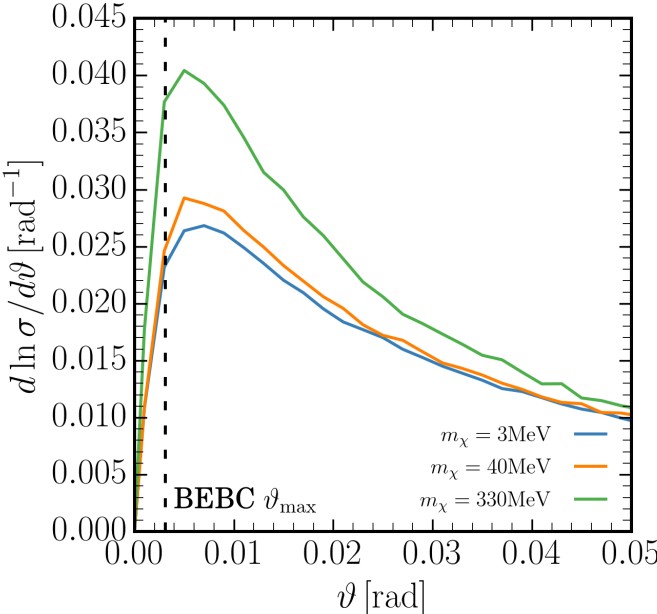

Figure 1: The differential angular distribution of dark states emerging from the beam dump.

mesons, the decay proceeds via an off-shell photon involving the interaction vertex dictated by the chiral anomaly

$$\mathcal{L}_{\mathfrak{s}} \supset \frac{1}{F_{\mathfrak{s}}} \mathfrak{s} F^{\mu\nu} \tilde{F}_{\mu\nu};$$  (6)

Note that the value of the decay constants here are irrelevant as we normalise to the observed Standard Model decay: $\mathfrak{s} \to \gamma\gamma$.

For dark states produced by vector meson decays, we invoke vector meson dominance, i.e. assume that the dominant interaction between vector mesons and photons occurs through mixing terms, which is in agreement with experimental data [37, 38]. Thus, the decays of vector mesons $\mathfrak{v}$ occur by mixing into an off-shell photon which can then decay into a dark state pair. To implement this into MADDUMP, we diagonalise the Lagrangian in the $(A^\mu, \mathfrak{v}^\mu)$ space. These two bases can be related by a series of two linear transformations, after which the original photon interactions of Eq.(1) result in three-point couplings between vector mesons and dark states, for instance

$$\mathcal{L}_{\mathfrak{v}} \supset c\epsilon e \mathfrak{v}^\mu \bar{\chi} \gamma_\mu \chi,$$  (7)

in the case of a millicharged particle; the constant $c$ depends on the couplings occurring in the original meson-photon mixing Lagrangian but their precise values are unimportant in practice once we normalise to the process $\mathfrak{v} \to e^+ e^-$.

After any dark states are produced, they propagate downstream of the dump through several hundred metres of material.[1] The geometric acceptance $\varepsilon_{\text{geo}}$ denotes the fraction of the dark states that then enter the detector. This is a function of both the angular distribution of the states as well as the angular size of the detector. Since the CERN SPS beam used by CHARM II and BEBC operated at high energies, most of the mesons produced in the dump had a large Lorentz boost $\Gamma$ in the forward direction. Going from the meson rest-frame to the lab-frame thus focusses the emitted dark states into a cone of opening angle $\vartheta \sim 1/\Gamma$. We find $\varepsilon_{\text{geo}} \sim 0.01$, which is much larger than the fractional solid angles of the detectors.

---

[1]It is theoretically possible that the states may interact strongly enough to be attenuated *en route* to the detector, although this possibility is ruled out in practice by constraints from other experiments.

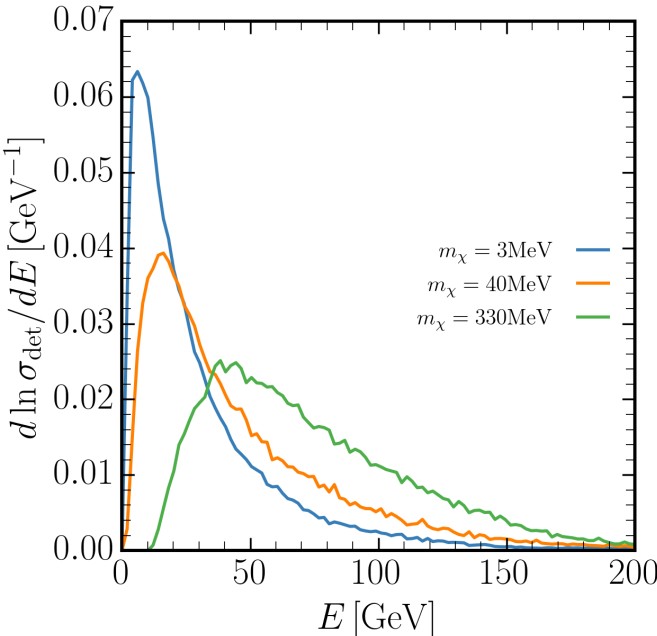

Figure 2: The differential energy distribution of dark states entering the BEBC detector.

## 3.2 Drell-Yan production

The dark states may also be produced by the Drell-Yan process. This is however subdominant to all the meson decays we consider, and so only becomes relevant when all other production processes are kinematically excluded, i.e. for $m_\chi$ above half the $J/\psi$ mass. We again model this using MADGRAPH through MADDUMP. Although states with masses beyond a few GeV can be produced in this way by the CERN SPS beam, the increase in sensitivity in this mass region is negligible. The scattering detection process we consider is only on electrons in the detector, as we detail in the next section. The GeV electron recoil energy thresholds of BEBC and CHARM II are too high to detect any scattering events of these heavy states, since the dark states do not have high enough Lorentz factors $\Gamma$ to deposit such large energies, with the maximum energy transfer scaling as $m_e \Gamma^2$. Consideration of deep inelastic scattering events does not change this conclusion.

## 3.3 Dark state-electron scattering

The dark states that enter the detector may either scatter via photon exchange off electrons or undergo deep inelastic scattering with the nucleons, however electron scattering dominates. We may write the total number of scattering events $N_{e\chi}$ as a function of final electron energy $E_e$ as

$$N_{e\chi} = \varepsilon_{\text{geo}} L n_e \int dE_\chi \frac{dN_\chi}{dE_\chi} \sigma_{e\chi}(E_\chi), \tag{8}$$

where $\varepsilon_{\text{geo}}$ is the geometric acceptance as defined in the previous section, $L$ is the longitudinal detector length, $n_e$ is the number density of electrons, $N_\chi$ is given by Eq.(2), and $\sigma_{e\chi}$ is the cross section for electron-chi scattering.

Due to the experimental cuts and their finite resolutions, not all of these events can be detected. We take this into account by counting the number of events that survive the cuts on

the electron angle with respect to the beam $\theta_e$ as well as on the electron energy $E_e$. The ratio of this number to $N_{e\chi}$ is denoted $\varepsilon_{\text{cut}}$, so that the total number of detected scattering events $N$ is given by:

$$N = \eta \varepsilon_{\text{cut}} N_{e\chi}, \tag{9}$$

where $\eta$ is the detector's efficiency. In practice, this scattering is handled by MADDUMP.

Cuts were applied on the electron energy $E$ and scattering angle $\theta$ of $E\theta^2 < 2m_e$ translating into a cut on the $t$-channel of approximately $t \lesssim -1 \times 10^{-3}$ GeV. This cut corresponds to the maximum expected scattering angle possible for incoming massless particles which are perfectly collimated along the beam axis, as was appropriate for neutrino experiments. This cut may be overzealous, as we find the non-zero spread of the incoming flux has on $\mathcal{O}(1)$ effect on the signal passing the selection cut at BEBC, even in the massless limit. For an electron with the minimum detected energy at BEBC of 1 GeV, the selection criterion $E_e\theta_e^2 < 2m_e$ means the scattering angle must satisfy $\theta_e \lesssim 0.03$ rad. Comparing this with the 9 mrad opening angle of the detector, we see that the detector angle is not negligible compared to the scattering angle cut, even at the low energy end of the tail. This in fact leads to about half of the signal events being thrown away.

## 4   Discussion

We now consider the bounds on the size of the electromagnetic coupling of dark states arising from the BEBC and CHARM II beam dump experiments. As already mentioned, a single elastically scattered electron was observed at BEBC. This event was likely due to neutrino electroweak scattering, which was carefully estimated to comprise a background of $0.5 \pm 0.1$ events [25]. The 90% CL upper limit on signal events is then 3.5.

The bounds from CHARM II are obtained by considering the sum of the observed electron events: $2677 \pm 82$ in the neutrino beam, and $2752 \pm 88$ in the anti-neutrino beam, making up $5429 \pm 120$ events. In the absence of any experimentally calibrated estimate of the background, we take the number of background events to be simply equal to the number of observed events. Assuming a Gaussian distribution, this places a 90% CL upper limit of 154 signal events. It may be that in fact the expected background is larger (smaller) than the number of observed events so the true bounds from CHARM II could be weaker (stronger) than those we find.

The bounds on millicharged particles coming from BEBC and CHARM II are shown in Fig. 3. The limits are improved on by subsequent experiments for masses below 100 MeV. However for heavier states, the higher energy of the CERN SPS beam becomes significant. The heavier mesons that are produced may decay into dark states of mass up to $\sim 1$ GeV, thus extending the reach by orders of magnitude. The two beam dumps have comparable sensitivities, although the combination of the lower energy threshold, larger angular size and lower backgrounds of BEBC allows it to probe somewhat deeper than CHARM II, notwithstanding the latter's much larger size.

For EDMs and MDMs, BEBC places the leading experimental bound and asymptotes to $d, \mu < 6.9 \times 10^{-6} \mu_B$ as shown in Fig. 4. The bounds tend to the same value for both operators, since in the relativistic limit the introduction of the $\gamma^5$ matrix in the EDM matrix elements leads only to a relative sign compared to MDM matrix elements, which is irrelevant for the observable here. At higher masses, there are fewer heavy mesons produced, while the high centre-of-mass energy of LEP has a larger role than in the SLAC mQ case. Hence the bounds we derive from BEBC become weaker than those from L3 at LEP II beyond a few hundred MeV.

The bound from CHARM II, which at low masses goes down to $d, \mu < 9.0 \times 10^{-6} \mu_B$, is slightly worse that the CHARM II bound of $\sim 8 \times 10^{-6} \mu_B$ found in previous work [23]. This may be explained by the combination of a number of factors: we find somewhat fewer dark

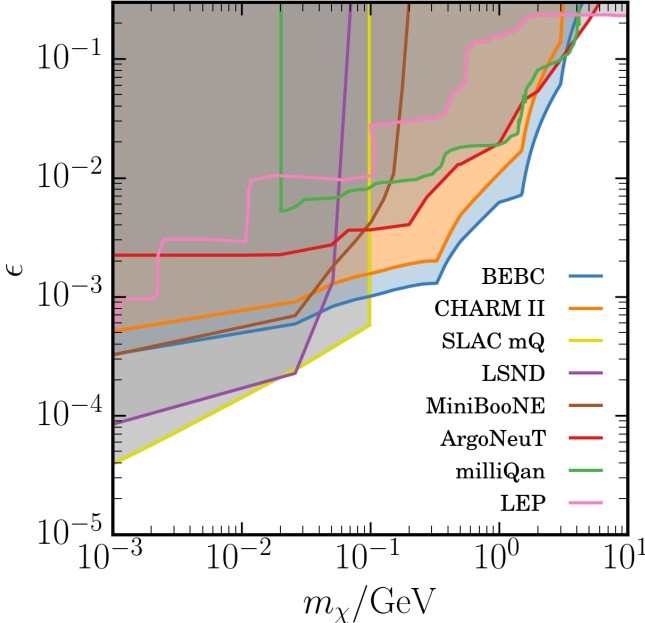

Figure 3: The 90% upper limit on the size of the millicharge $\epsilon = Q_\chi/e$ from CHARM II and BEBC. All regions shaded in grey are already excluded at 90% by: SLAC [4]; LSND and MiniBoone [7]; ArgoNeuT [9]; milliQan [10]; and LEP [12].

states enter the detector due to our different method of treating the meson production, as explained in § 3.1; the true CHARM II cut we use is somewhat more restrictive than that used in [23]; we use a lower electron detection efficiency; additionally, as mentioned at the end of § 3.3, the finite angular size of the dark state flux cone can have $\mathcal{O}(10\%)$ effects.

We have shown that the BEBC WA66 beam dump experiment [24] carried out in 1982, which was previously used for a number of novel searches [25–27], continues to place world-leading bounds on several 'dark currents' coupling to photons. This lends further support to the proposal to reexamine neutrino data in the search for new dark states [39]. We expect that similar improved bounds may be placed using BEBC data on other feebly interacting particles of current interest [3], in particular heavy neutral leptons [40].

# Acknowledgments

We are very grateful to Wilbur Venus for helpful discussions and encouragement. We thank Luca Buonocore, Xiaoyong Chu, Claudia Frugiuele and Matthieu Marinangeli for much useful correspondence. We also thank Zhen Liu and Roni Harnik for informing us about related work. This paper is dedicated to the memory of Per-Olof Hulth, Spokesperson of the CERN-WA-066 BEBC beam dump experiment, which continues to provide such rich yields nearly four decades later. And thanks to all his colleagues without whom this experiment would not have happened.

# A    Details of meson decays

As mentioned in the main body of the text, the normalisation of the number of dark states coming from meson decays is set by the corresponding branching ratio: $\mathrm{Br}(\mathfrak{s} \to \chi\bar{\chi}\gamma)$ for

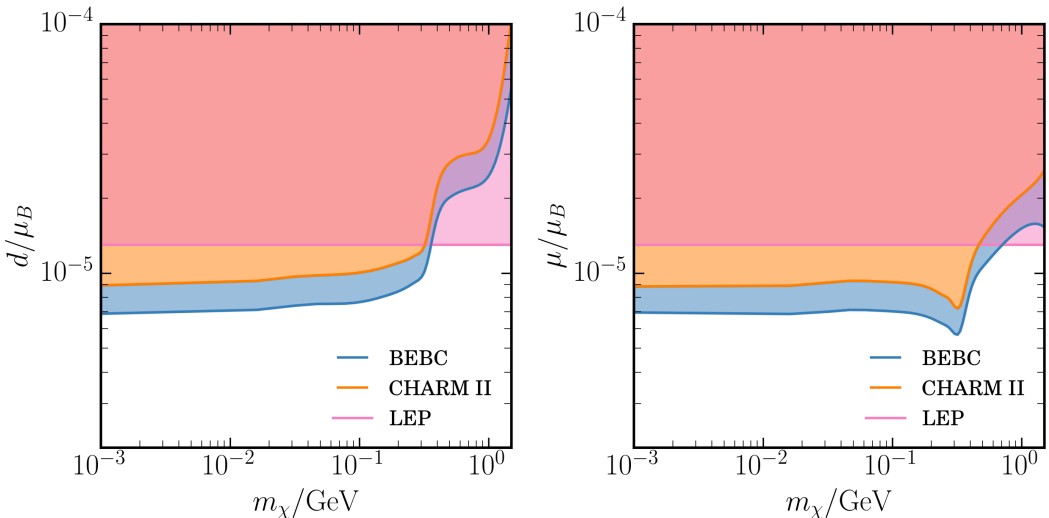

Figure 4: The 90% upper limit exclusion regions for electric $d$ (left) and magnetic $\mu$ (right) dipole moments, measured in Bohr magnetons $\mu_B$. The bounds from L3 at LEP II [22] are also shown. The CHARM II bounds we derive here agree with those found earlier [23] and are less restrictive than those of BEBC.

scalar mesons, $\mathrm{Br}(\mathfrak{v} \to \chi \bar{\chi})$ for the vectors, as well as $\mathrm{Br}(\omega \to \pi^0 \chi \bar{\chi})$. In this Appendix, we calculate these rates and reproduce for completeness many of the relevant key formulae of Refs. [23,41], to which we refer the reader for further details.

The branching ratio for dark decays of scalar mesons can be related to the known branching ratio of some Standard Model process. In general, the ratio of the branching ratios is given by the ratio of the corresponding rates. The simplest ratio to calculate in this case is dark decay relative to the decay into two photons, i.e.

$$\mathrm{Br}(\mathfrak{s} \to \chi \bar{\chi} \gamma) = \frac{\Gamma(\mathfrak{s} \to \chi \bar{\chi} \gamma)}{\Gamma(\mathfrak{s} \to \gamma \gamma)} \mathrm{Br}(\mathfrak{s} \to \gamma \gamma). \tag{10}$$

This ratio factorises nicely if the momentum-transfer-dependence of the meson electromagnetic form factors is neglected. The expressions for the decay rates are [41]

$$\Gamma(\mathfrak{s} \to \chi \bar{\chi} \gamma) = \int_{4m_\chi^2}^{m_\mathfrak{s}^2} ds_{\chi\bar{\chi}} \, \Gamma_{\gamma\gamma^*}(s_{\chi\bar{\chi}}) \frac{f_\chi(s_{\chi\bar{\chi}})}{16\pi^2 s_{\chi\bar{\chi}}^2} \sqrt{1 - \frac{4m_\chi^2}{s_{\chi\bar{\chi}}}}, \tag{11}$$

where $m_\mathfrak{s}$ is the scalar meson mass and

$$\Gamma_{\gamma\gamma^*}(s_{\chi\bar{\chi}}) = \frac{\alpha^2 \left(m_\mathfrak{s}^2 - s_{\chi\bar{\chi}}\right)^3}{32\pi^3 m_\mathfrak{s}^3 F_\mathfrak{s}^2}, \tag{12}$$

is the decay rate of a scalar meson to two photons, one real and one of virtuality $s_{\chi\bar{\chi}}$; then $\Gamma(\mathfrak{s} \to \gamma\gamma) = \Gamma_{\gamma\gamma^*}(0)$. Note that the final branching ratio is independent of the meson decay constants $F_\mathfrak{s}$ in this approximation. The expressions for $f_\chi(s)$ depend on the particular

interaction term being considered, and were calculated in [41] to be

$$\text{mQ} : f_\chi(s) = \frac{16\pi\alpha}{3}\epsilon^2 s \left(1 + \frac{2m_\chi^2}{s}\right),$$

$$\text{MDM} : f_\chi(s) = \frac{2}{3}\mu^2 s^2 \left(1 + \frac{8m_\chi^2}{s}\right), \tag{13}$$

$$\text{EDM} : f_\chi(s) = \frac{2}{3}d^2 s^2 \left(1 - \frac{4m_\chi^2}{s}\right).$$

The vector meson branching ratio into pure dark states is obtained similarly, and is most simply given by

$$\text{Br}(\mathfrak{v} \to \chi\bar{\chi}) = \frac{\Gamma(\mathfrak{v} \to \chi\bar{\chi})}{\Gamma(\mathfrak{v} \to e^- e^+)}\text{Br}(\mathfrak{v} \to e^- e^+). \tag{14}$$

Under the vector meson dominance hypothesis, the mixing terms between the vectors and the photon imply that any terms in the Lagrangian involving the "original" non-diagonalised photon field in fact involve some linear combination of the diagonal fields. Hence Eq. (1) gives rise to a direct interaction between the diagonalised vector meson and the dark states. Both of the decays in Eq.(14) have just two-body final states so the phase spaces contributions factorise, leaving

$$\frac{\Gamma(\mathfrak{v} \to \chi\bar{\chi})}{\Gamma(\mathfrak{v} \to e^- e^+)} = \frac{f_\chi(m_\mathfrak{v}^2)}{f_e(m_\mathfrak{v}^2)}\sqrt{\frac{1 - 4m_\chi^2/m_\mathfrak{v}^2}{1 - 4m_e^2/m_\mathfrak{v}^2}}, \tag{15}$$

where $f_e(s)$ is analogous to the millicharge $f_\chi$:

$$f_e(m_\mathfrak{v}^2) = \frac{16\pi\alpha}{3}m_\mathfrak{v}^2\left(1 + \frac{2m_e^2}{m_\mathfrak{v}^2}\right). \tag{16}$$

The final branching ratio concerns the decay of a vector meson into a pion and a dark pair, which we normalise to the branching ratio into a pion and photon:

$$\text{Br}(\mathfrak{v} \to \chi\bar{\chi}\pi^0) = \frac{\Gamma(\mathfrak{v} \to \chi\bar{\chi}\pi^0)}{\Gamma(\mathfrak{v} \to \pi^0\gamma)}\text{Br}(\mathfrak{v} \to \pi^0\gamma). \tag{17}$$

The relevant interactions here come from the original chiral anomaly term coupling the pion to $\mathfrak{s}F\tilde{F}$. Diagonalisation turns this interaction into a sum of two new interactions: a term involving a vector meson and a photon, and a term involving two vector mesons. Assuming the mixing terms are sufficiently weak, we may, to leading order, consider only the interaction involving a photon and vector meson, which we show below to be valid. The decay rate for this process is then:

$$\Gamma(\mathfrak{v} \to \chi\bar{\chi}\pi^0) = \int_{4m_\chi^2}^{(m_\mathfrak{v} - m_\pi)^2} ds_{\chi\bar{\chi}}\, \Gamma_{\pi^0\gamma^*}(s_{\chi\bar{\chi}})\frac{f_\chi(s_{\chi\bar{\chi}})}{16\pi^2 s_{\chi\bar{\chi}}^2}\sqrt{1 - \frac{4m_\chi^2}{s_{\chi\bar{\chi}}}}, \tag{18}$$

where

$$\frac{\Gamma_{\pi^0\gamma^*}(s_{\chi\bar{\chi}})}{\Gamma(\mathfrak{v} \to \pi^0\gamma)} = \frac{m_\mathfrak{v}^2(m_\pi^2 - m_\mathfrak{v}^2 - s_{\chi\bar{\chi}})^2}{(m_\mathfrak{v}^2 - m_\pi^2)^3}\sqrt{1 - \frac{2(m_\pi^2 + s_{\chi\bar{\chi}})}{m_\mathfrak{v}^2} + \frac{(s_{\chi\bar{\chi}} - m_\pi^2)^2}{m_\mathfrak{v}^4}}, \tag{19}$$

is the rate of vector meson decay into a pion and photon of virtuality $s_{\chi\bar{\chi}}$ compared to the corresponding on-shell rate, and $m_\mathfrak{v}$ and $m_\pi$ are the vector meson and pion mass, respectively. As a check on the weak-mixing assumption, we use the above expression to find

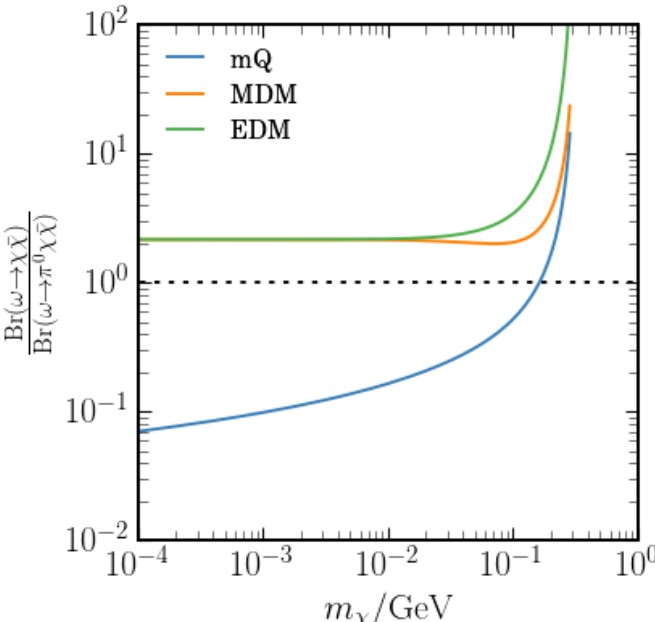

Figure 5: The ratio of the two branching ratios for the $\omega$ meson's two main decay channels involving a dark pair, as given by equations (14) and (17). For the dimension-5 operators, MDM and EDM, the decay involving a pion is always at least marginally subdominant, whereas for mQ there are dark state masses for which the 3-body decay is dominant.

$\mathrm{Br}(\omega \to \pi^0 e^+ e^-) = 8.3 \times 10^{-4}$ and $\mathrm{Br}(\omega \to \pi^0 \mu^+ \mu^-) = 1.3 \times 10^{-4}$, both of which are within 10% of their experimental value.

The relative importance of this decay channel compared to the decay without a pion depends on the particular form of the interaction as well as the value of $m_\chi$ (see Fig. 5). EDMs and MDMs are higher dimension operators than the standard electromagnetic current, resulting in a stronger energy dependence. The reduced phase space associated with the decay into a dark pair and a pion then has much more of an effect on particles coupling through the former operators, so in such cases we can consider this channel to be negligible.

However for low mass millicharges, the decay channel involving a pion is dominant. At very low mass, pion decay is the dominant production mode, but at higher masse when this channel starts to shut off, the inclusion of $\omega \to \chi \bar{\chi} \pi^0$ can make $\sim 5\%$ difference to the bounds. At even higher mass, this $\omega$ decay channel becomes negligible but including $\omega \to \chi \bar{\chi}$ yields $\sim 10\%$ improvement. Hence we include both channels to accurately cover the whole range of masses.

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
