# Peer review of "Blast from the past: Constraints on the dark sector from the BEBC WA66 beam dump experiment"

_SciPost Physics, doi:SciPost Phys. 10, 043 (2021)_

## Round 1 · Referee Report · Anonymous (Referee 1) · 2020-12-22

Report
The paper considers the production of a pair of new Dirac fermions
chi through their coupling to the Standard Model photon either via
millicharge or through magnetic/electric dipole moments in fixed
target proton beam experiments. Decay yields from scalar and vector
mesons as well as Drell-Yan production are modeled using the
MadGraph plugin MadDump. The total number of scattering events in a
downstream detector is then estimated. Old data from the CERN BEBC
experiment are utilized to set the most stringent direct
constraints on the size of the aforementioned interactions.
The paper is a welcome and timely addition to the general ongoing
"intensity-frontier" efforts that seek to chart out the sub-GeV
landscape of new dark states coupled to Standard Model
particles. The paper is well-written, clear in its physics and I am
happy to recommend publication after the following remarks have been
addressed:
The authors make a point about the difference between charged
and neutral pion decay distributions, owed to their markedly
different lifetime and subsequent differing interactions with the
target. However, it did not become clear to me to what extend this
is actually taken into account? MadDump is used, but it appears its
primary purpose is to infer the angular and energy differential chi
distributions given the initial meson spectra? But what is assumed
for the latter? Finally, plots of the chi-distribution in angle and
energy would be helpful.
chi through their coupling to the Standard Model photon either via
millicharge or through magnetic/electric dipole moments in fixed
target proton beam experiments. Decay yields from scalar and vector
mesons as well as Drell-Yan production are modeled using the
MadGraph plugin MadDump. The total number of scattering events in a
downstream detector is then estimated. Old data from the CERN BEBC
experiment are utilized to set the most stringent direct
constraints on the size of the aforementioned interactions.
The paper is a welcome and timely addition to the general ongoing
"intensity-frontier" efforts that seek to chart out the sub-GeV
landscape of new dark states coupled to Standard Model
particles. The paper is well-written, clear in its physics and I am
happy to recommend publication after the following remarks have been
addressed:
The authors make a point about the difference between charged
and neutral pion decay distributions, owed to their markedly
different lifetime and subsequent differing interactions with the
target. However, it did not become clear to me to what extend this
is actually taken into account? MadDump is used, but it appears its
primary purpose is to infer the angular and energy differential chi
distributions given the initial meson spectra? But what is assumed
for the latter? Finally, plots of the chi-distribution in angle and
energy would be helpful.

---

## Round 2 · Author Response

"The authors make a point about the difference between charged and neutral pion decay distributions, owed to their markedly different lifetime and subsequent differing interactions with the target. However, it did not become clear to me to what extent this is actually taken into account? MadDump is used, but it appears its primary purpose is to infer the angular and energy differential chi distributions given the initial meson spectra? But what is assumed for the latter?
We model the initial meson spectra as proton-proton collisions scaled according to the nucleon number of the target nuclei; concretely, the scaling goes as A^{2/3}. We are aware of the subtleties hinted at by the Reviewer but in actual fact made a crude approximation since it makes little difference to our final result. Indeed (as was pointed out to us by W. Venus, private communication), there are two corrections to our results which are of *opposite* sign. That we ignore the contribution from the nuclear cascade makes our bounds conservative, whereas that we also ignore “reinteractions in the nucleus” (i.e. slowing down in nuclear matter of the quarks and gluons off which the DM particles are created) makes them less so. However these effects are at most 20-30% in the rates and probably cancel out - so the net effect would not be discernible on our final limit which goes as the 4th root of the rate.
"Finally, plots of the chi-distribution in angle and energy would be helpful."
We include a plot of 1/\sigma d\sigma/dE versus E for DM particles entering BEBC, coming from all channels for 3 nominal values of DM mass (3, 40 and 330 MeV). Also a plot of 1/\sigma d\sigma/d\theta against \theta for all the DM particles being produced in the beam dump, again for 3 DM masses (3, 40 and 330 MeV).
The curves are jagged because of limited statistics in running the simulations. We hope the overall trends reassures the Reviewer. Would they like us to keep these in a new version of the paper? (In that case we will probably need to run more simulations to make the figures look prettier.)
We model the initial meson spectra as proton-proton collisions scaled according to the nucleon number of the target nuclei; concretely, the scaling goes as A^{2/3}. We are aware of the subtleties hinted at by the Reviewer but in actual fact made a crude approximation since it makes little difference to our final result. Indeed (as was pointed out to us by W. Venus, private communication), there are two corrections to our results which are of *opposite* sign. That we ignore the contribution from the nuclear cascade makes our bounds conservative, whereas that we also ignore “reinteractions in the nucleus” (i.e. slowing down in nuclear matter of the quarks and gluons off which the DM particles are created) makes them less so. However these effects are at most 20-30% in the rates and probably cancel out - so the net effect would not be discernible on our final limit which goes as the 4th root of the rate.
"Finally, plots of the chi-distribution in angle and energy would be helpful."
We include a plot of 1/\sigma d\sigma/dE versus E for DM particles entering BEBC, coming from all channels for 3 nominal values of DM mass (3, 40 and 330 MeV). Also a plot of 1/\sigma d\sigma/d\theta against \theta for all the DM particles being produced in the beam dump, again for 3 DM masses (3, 40 and 330 MeV).
The curves are jagged because of limited statistics in running the simulations. We hope the overall trends reassures the Reviewer. Would they like us to keep these in a new version of the paper? (In that case we will probably need to run more simulations to make the figures look prettier.)

---

## Round 2 · List of Changes

Added plots of differential distribution of dark states in angle and energy. Added comment on this at end of page 4.

---

## Editorial Decision

published